# Estimating Discard Mortality in Commercial Fisheries without Fish Dying: A 3R Challenge

**DOI:** 10.3390/ani12060782

**Published:** 2022-03-19

**Authors:** Niels Madsen, Rasmus Ern, Aage Kristian Olsen Alstrup

**Affiliations:** 1Department of Chemistry and Bioscience, Aalborg University, 9220 Aalborg, Denmark; rasmus@ern.dk; 2Department of Nuclear Medicine and PET Centre, Aarhus University and Hospital, 8200 Aarhus, Denmark; aage.olsen@clin.au.dk

**Keywords:** 3R, discard mortality, E.U. landing obligation, fisheries management, fish welfare, laboratory animal legislation

## Abstract

**Simple Summary:**

Due to the implementation of a landing obligation in fisheries in the European Union (EU), with an exemption for species with “high survival”, discard survival studies (henceforth DSSs) have become one of the most politically prioritized fisheries research areas in European fisheries. In contrast to most fisheries science research areas, DSSs are embedded by the EUs animal research welfare directive. This is a challenge, and there has not been any focus on how to implement animal welfare (AW) regulations experimentally in DSSs. This paper focuses on AW regulations in relation to conducting DSSs, but the outreach is much broader. We investigate experimental procedures by bringing in relevant examples, using the output results, and relating this information to relevant AW guidelines and regulations by focusing on implementing 3R principles.

**Abstract:**

Globally, it is estimated that around 10% of the fish that are caught are discarded. This is considered to be a wasteful human marine activity since these fish are often dead or dying. To reduce the high discard rates of commercial fisheries, the European Union (E.U.) has enacted a landing obligation that includes the ability to exempt “species for which scientific evidence demonstrates high survival rates”. Therefore, discard survival studies (henceforth DSSs) have become one of the most politically prioritized fisheries research areas in European fisheries. International expert groups have produced guidance reports to promote best practices and to harmonize the methodologies. Nevertheless, there has not been any focus on how to implement animal welfare (AW) regulations experimentally. Discard survival studies are “frontrunners” in fisheries science research areas that are embedded by animal research welfare requirements and are expected to be more restrictive in the future because of an increased public focus on fish welfare. This paper focuses on AW regulations in relation to conducting DSSs, but the outreach is much broader. We investigate experimental procedures by bringing in relevant examples, using output results, and relating this information to relevant AW guidelines and regulations by focusing on implementing 3R principles.

## 1. Introduction

### 1.1. The European Union Landing Obligation and the Exemption

The catches from commercial fisheries are an important protein resource for human consumption around the world. Although global fish catches have nearly reached their maximum capacity [1,2], it is estimated that around 10% of global catches are discarded, with these discards being considered wasteful human marine activities [1,3]. Discards refer to the organisms of both commercial and non-commercial value that are caught during commercial fishing operations and that are returned to the sea, often dead or in dying condition [4]. In the European Union (E.U.), fisheries discard rates are also high, even for commercially important fish species where they can exceed landings [5]. The E.U. considers discards to be a substantial waste of resources and believe that they negatively affect the sustainable exploitation of marine biological resources and marine ecosystems as well as the financial viability of fisheries. Consequently, the introduction of the E.U. (E.U.) Common Fisheries Policy (CFP) resulted in a landing obligation that has been in effect since 2015, which prohibits the discard of fish species that are regulated by a quota (E.U. regulation, 1380/2013). Since there are several commercially important fish species that are likely to survive being discarded and can contribute to maintaining the population, the landing obligation therefore includes the possibility to exempt “*species for which scientific evidence demonstrates high survival rates, taking into account the characteristics of the gear, of the fishing practices and of the ecosystem*” (E.U. regulation, 1380/2013, Article 15, paragraph 4b). With this change, the E.U. CFP sparked a sudden and uniquely synchronized interest in research to demonstrate “high survival rates” in many European fisheries [6].

### 1.2. Discard Survival Studies: Scientific Evidence

To provide scientific evidence for the “high survival exemption”, this research area has become one of the most politically prioritized fisheries research areas in European fisheries in recent years. This research area has developed rapidly, and the ICES (The International Council for the Exploration of the Sea) established the ICES Working Group on Methods for Estimating Discard Survival (WGMEDS) [6,7,8]. The WGMEDS authored a comprehensive guideline report (223 pages) “ICES Guidelines on Methods for Estimating Discard Survival” [8]. However, the guidelines do not address how to handle animal welfare (henceforth AW) regulations, despite this is becoming of increasing importance when planning discard survival studies (henceforth called DSSs). The guideline report mentions that, in a few lines at the end (p. 169), it might be necessary to apply for licenses, and applications can be time consuming [8]. Consequently, there is no guidance for researchers conducting DSSs, although all experiments that are related to the European landing obligation are embraced by the same E.U. regulations (Directive 2010/63/EU). Additionally, a focus on AW in relation to fish (and other aquatic animals) is increasing [9,10,11,12] and unquestionably has an increasingly influential effect on research activities, with DSSs being among the most exposed research areas related to commercial fisheries. The reasons for these changing perceptions are due to findings related to indications of neurophysiological pain-related similarities (nociception) [13,14,15] and cognitive abilities [16] to mammals that conflicts with the earlier views of fish as inferior and non-feeling vertebrates [17]. Additionally, fish are increasingly being used as model species in animal experiments (specifically, the Zebra fish is often used in basic and medical research), and there is public concern about fish welfare in the growing fish farming industry [10,13], as well as about fish welfare in commercial fisheries [18,19]. Additionally, scientific journals have increased their focus on AW, require ethical statements and have policies related to AW, and it is essential to consider AW regulations and requirements when planning the experiments.

### 1.3. Animal Welfare: Challenges Experienced by Researchers

This issue has left many researchers (including the authors) on new and challenging ground. The majority of researchers joining this research field have comprehensive research experience related to commercial fisheries, where large amounts of fish are handled and measured on the deck of a fishing vessel and often die without any euthanization (intentionally ending life to relieve pain and suffering), and no research activities related to AW regulations. Researchers often lack basic knowledge and experience on how to keep and handle living animals in laboratories as well as knowledge on the physiological status of individual fish. Incorporating AW regulations in these experiments takes extra effort in terms of planning and increases resource use (increasing budgets).

DSSs are “frontrunners”, but far more research areas related to commercial fisheries will be affected by the trend of introducing more restrictions related to AW. Therefore, this paper is also relevant for researchers working in other areas of fisheries research as well as all relevant stakeholders in relation to DSSs with limited AW knowledge. These stakeholders include the administrators of research funding agencies and evaluators of research proposals, the politicians responsible for regulations, the mangers evaluating results in relation to the “high survival exemption”, the reviewers and editors of scientific journals, and the fishing industry.

Although AW is an increasing issue that can have a substantial effect on fisheries research projects, there has been very limited scientific focus in terms of how to handle this. In the present study, we investigate experimental procedures by bringing in relevant examples, using the output results, and relating this information to relevant AW guidelines and regulations. We also consider how relevant issues can be addressed using the experimental approach introduced here. Our approach is pragmatic and focuses on the scientific aspect of conducting experiments, recognizing that fundamental ethical regulations might not always be logical for a researcher, but that they are present and will increasingly continue to influence our research activities. Although are focusing on regulations in the European Union, these are built on internationally established principles, particularly the 3Rs (replacement, reduction, and refinement).

A glossary is provided in Appendix A.

## 2. DSS Methods, Evaluation Procedures, and a Case Study

### 2.1. Aims and Basic Methods

The overall aim of fish DSSs is to estimate discard survival for a specific species in a specific fishery to provide evidence for potentially “high survival”. Basically, they can be described as a binomial process where the outcomes can be classified in only one of two mutually exclusive ways: dead or alive. DSSs are generally designed to be representable for the fishery assessed. This implies the use of commercial fishing vessels as well as commercial fishing and handling practices.

The methodology that is the most used by far, and therefore our primarily focus, is captative observations where the fish are stored and observed in over a period of time, though tagging is also used [8,20,21], and vitality assessments [22,23,24,25] are implemented for proxy survival estimates. Typically, the fish are collected on fishing vessels carrying out commercial fishing operations and are stored in deck tanks and brought to observation sites. The majority of discard mortality studies house fish in land-based holding tanks [26,27,28,29,30,31,32,33,34,35]. Alternatively, the fish can be held in livewells [36] or cages that are anchored to the seafloor [37]. More generally, capitative experiments can be characterized as short-term survival since observation periods are usually less than one month, with the majority of studies lasting for less than two weeks and most only lasting just a few days (examples in Table 1).

### 2.2. Scientific Evaluation of DSSs

The Scientific, Technical and Economic Committee for Fisheries (STECF) performs scientific work that is directly requested by the E.U. Commission and is the advisory body that provides a systematic and consistent evaluation process. However, assessing the robustness of evidence and defining “*high survival*” has proven to be challenging for the STECF [6]. At least 20 species from the Mediterranean to the Baltic seas have been assessed [6]. These evaluations considered whether the survival assessment methods are appropriate and whether the limitations of the results have been fully explored [6]. The ICES guidance on survival assessment [7,8,38] protocols has been used to promote best practices and the harmonization of the more recently conducted DSSs. No clear guidance has been provided by STECF, but many DSSs studies have been criticized for lacking sufficient data in the data presented in [6]. The STECF does not consider AW issues. In practice, grant exceptions from the E.U. landing obligation based on discard survival estimates range from 46 to 90% [34].

### 2.3. Case Study: Plaice DSSs

To concretize and analyze methodology approaches in DSSs in relation to AW regulations, we conducted a case study by collecting and analyzing information concerning experimental approaches and results from previously published (peer reviewed) DSSs (Table 1). We chose plaice (*Pleuronectes platessa*) from DSSs conducted during the winter (regulations are often implemented seasonally). Plaice is likely the best example since it is the most studied fish species, and several landing obligation exemptions have been granted for plaice fisheries, with most being implemented for winter fishing. Seven studies from different areas are presented in Table 1, and data were collected using different types of commercially fished trawls (Table 1). Most of them have been used in other STECF evaluations [36].

It is notable that even studies with the same overall objectives show a marked variation in the estimated survival (15 to 89%) and in their experimental methodology (Table 1). In six of the seven studies, fish are stored in land-based holding systems, and in a single study, they are placed in cages placed on the seabed (Table 1). Marked variations can be observed in number of haul replicates (5–17), in the duration of the observation periods (3–21 days), and in the numbers of assessed (54–349) and surviving plaice (41–266). Of importance in relation to AW regulations is when a fish dies during the observation studies. We analyzed this and found that most studies (6 out of 7) observe the highest survival within the initial 0–3 days. The period without mortality is 1–3 days, before the end of the observation period, meaning that mortality is still observed close to experimental closure. In four studies, mortality above five percentage points was still able to be observed during the last three days of the observation period. In several assessments, a vitality assessment is conducted before the start of the observation period (after capture). Fish in captivity are inspected from zero-to-three times per day.

## 3. Fish Welfare Regulations and Scientific Journal Requirements

### 3.1. Scientific Journal Requirements

Most international scientific journals, the medium through which DSSs are published, require that the work described in published articles is carried out in accordance with specified ethical guidelines and request further author declarations attesting compliance with those guidelines. However, scientific journals do not necessarily refer to the same collection of guidelines, even though there might not be a marked variation in their basic principles. Many journals require that the name of the licensing authority and the license number be disclosed in the manuscript. A national permission from authorities is not necessarily sufficient to fulfill the ethical requirements of certain journals. Therefore, it is important to consider the potential ethical requirements of a journal before setting up experiments. The following is a list of relevant guidelines that some journals refer to: “*The Code of Ethics of the World Medical Association (Declaration of Helsinki) for animal experiments*” (www.wma.net; accessed on 16 March 2022)*,* the ARRIVE guidelines (Animal Research: Reporting of in vivo Experiments) (https://arriveguidelines.org/; accessed on 16 March 2022), and the European Union directives described in more detail below (Directive 2010/63/EU). Many journals also require the name of the national institution that grants approval for the use of animals in the research and the reference number for animal ethics approval. With the increased use of online journals, it is often possible to make use of supplementary materials to account for ethical aspects and for the documentation of compliance guidelines. For the ARRIVE guidelines, journals often demand that a form documenting where the desired information is stated in the manuscript is completed.

### 3.2. Legislation

Globally, the legislation on protected experimental animals differs between countries and geographical regions [39]. In several countries, all animals are, in principle, protected; in some countries, only warm-blooded vertebrates are protected, while in other places, all invertebrates (including amphibians and fish) and cephalopods are protected (Europe is included in this category). Some countries protect animals at all stages of development, while others are only protected from defined early stages or as adults [39].

The constitutional basis of the European Union stipulates that, as sentient beings, animals should have their welfare requirements paid full regard (Treaty of Lisbon, Article 13, Official Journal of European Union, 2007/C 306/01). Animals must not be used for scientific research in the E.U. unless there is justification, in which case the expected benefits outweigh animal suffering (the so-called beneficial criteria) or when the study’s objectives cannot be achieved using alternative methods that do not include living animals. The use of animals in research must follow E.U. directives, which are implanted in the national laws of the member states. The first E.U. regulations concerning the protection of animals used for scientific experimental purposes were published in 1986 (Directive 86/609/EEC). These regulations were updated in 2010 to improve AW through the implementation of more stringent and transparent measures, and to increase the minimum standards based on new scientific knowledge on the capacity of animals to sense and express pain, suffering, distress, and lasting harm. We extracted some of the most essential parts related to DSSs (and fisheries) dictated by Directive 2010/63/EU, and these are presented in Table 2. As is the case with all other vertebrates, fish are included in the directive (Table 2, A). All DSSs related to the European discard ban are therefore subjected to the same minimum regulations, but certain nations may have more extensive AW regulations (Table 2, C). Therefore, there is variability in compliance between studies.

European Union regulations are built on internationally established principles that are primarily founded on the principles of the 3Rs (replacement, reduction, and refinement) (Table 2, E). Therefore, the implementation of the 3R principles form the core of research design. The 3R principles were developed by William Russel and Rex Burch over 60 years ago in one of the most fundamental books on laboratory animal science: *The principles of humane experimental technique* [40] (see also Balls and Parascandola [41]). The originally proposed 3Rs, which reflected the order in which they should be addressed, were defined as [42] replacement: the substitution of conscious, living higher animals with insentient material; reduction: the number of animals used to obtain a certain amount of information and precision should be reduced; and refinement: any remaining severity in the inhumane procedures applied to study animals that still needs to be used should be decreased. For researchers, the 3R concept is therefore dynamic and conceptual since it is built on active and continuous improvements, which highlights the necessity of sharing and searching for new information as well as for continuous learning and reflection among researchers. For this reason, several national states have established 3R centers, the purpose of which are to disseminate knowledge of the 3R principles and to direct contact between researchers to direct concrete proposals for their practical implementation.

Animal research is generally defined by the needle criteria (Table 2, B), which is relevant for DSSs where fish are tagged, for individual recognition, or when using needle injection when collecting blood samples (for example [29]), but these criteria are also relevant in studies for which more invasive handling methods are used. A major challenge is the potential interpretation of DSSs as having a terminal endpoint (Table 2, H), which conflicts with the aim of observing mortality. This is an obvious paradox for researchers since, in principle, this fish fate is identical to what the fish would experience after discard from a commercial fishing vessel. Additionally, the alternative practice, under a discard-banned regime, would often be that the “discard fish” would die on deck since smaller fish under a minimum reference size are generally not euthanized. Nevertheless, when DSSs are authorized, it is because they ultimately benefit depleted fish stocks and, hence, the environment and a better use of limited human food resources (Table 2, G).

The administration of the E.U. directive might differ between countries (Table 2, C), but, generally, experiments must be authorized under the national regulations and licenses that are provided and that are authorized by committees. Some national animal research welfare committees have a legal requirement to involve stakeholders other than researchers in the evaluation of research proposals, including delegates from animal rights groups, theologians, and philosophers [9]. As an example, The Animal Experimental Council in Denmark (www.foedevarestyrelsen.dk/english/Animal/AnimalWelfare/The-Animal-Experiments-Inspectorate/Pages/default.aspx; accessed on 16 March 2022) consists of eleven experts from relevant subject areas that have been appointed by the Minister of Environment and Food; the council comprises one representative from each field: science, the health authorities, the Danish Industry, disease-fighting NGOs, and The Danish Animal Ethics Council, and there are four members from AW organizations. Therefore, the panel has a diverse range of expertise, but there is potential diversity regarding each group’s views on the E.U. directive. The evaluation of research proposals should be timely; therefore, it is recommended that researchers collect information on their proposed practices prior to application. The required public availability (Table 2, R) of information pertaining to granted permissions is an important source of information for researchers, and therefore it is also important to indicate the permission number in publications.

## 4. Considerations of Implementation of the 3R Principles in DSSs

### 4.1. Replacement: Methods to Avoid or Replace the Use of Fish

A clearer definition of “scientific evidence” that is sufficient to grant exemptions from the discard ban is essential for replacement methods. For example, a modification to the paragraph in the direction of “lack of scientific evidence for high mortality” could eliminate the necessity of conducting experiments in certain cases where scientific information is already available. A STECF guidance document collating the available information and describing the relevance and necessity of new DSSs would be useful. The level of scientific evidence-based information continues to increase as new studies are conducted, and a broader use of information across studies reduces the number of necessary experiments. Some recent studies are not fully publicly available, although they are assessed by STECF, and a structured environment in which information from past experiments can be shared would reduce the numbers of conducted experiments. A relevant classification system across European fisheries, including essential and already-known influential key factors, such as species, fishing gear category, season (temperature), and air exposure (catch handling time), could also be established. There is evidence-based information that identifies the parameters that influence survival, such as air exposure [29,30,43], capture depth [44], temperature [33,34,37,44], and phylogeny [44]. Any combination of these factors would exclude several fisheries and might be a more efficient way to build up a scientific framework and limit the necessity of new DSSs with a narrower scope. Additionally, broader reviews collecting and analyzing information from a larger number of studies [20,21,44,45] can prove to be efficient in providing information to identify new and relevant DSSs.

The use of vitality assessments is a widely recognized methodology [8,23,25,43,44] that can be used to indirectly predict species-specific discard survival from validated indicators [8]. Vitality indicators include reflex impairments and injuries, which can be assessed using the reflex action mortality predictors (RAMPs) method [22,23,24] and the catch damage index (CDI) [25]. The RAMP method tests the ability of fish to interact with the surrounding environment; consists of a suite of species-specific observations of voluntary behavior, response to stimuli, and clinical reflexes [28,46]; and has been used on a number of teleost species. RAMP has been identified as a robust metric method for predicting fish survival, with ample training for scientific staff using this rating scale and choice of reflexes that can be given objective scores [47]. Vitality assessments can also be conducted on fishing vessels efficiently, with the necessary handling usually not conflicting with the needle criteria (Table 2, B) while also providing satisfactory results, causing limited pain (Table 2, D), and, most importantly, avoiding death as an endpoint (Table 2, H). It is important that RAMP candidates are validated, and in some cases, they can be reduced to only three [32], allowing for the assessment to be carried out in less than one minute. In cases where the harm exceeds the needle criteria, the fish can be euthanized immediately after the RAMP assessment (Table 2, K, J). There are also some experimental advantages for captative DSSs due to a potentially higher coverage of the fishing fleet and far higher number of fish assessed.

Accepting validated vitality studies as “scientific evidence” would diminish the necessity of captative DSSs. The use of vitality assessments using verified candidates has the potential replace DSSs in relation to the initial assessment of unstudied fish species and fisheries before setting up DSSs; broader validation for specific species (for instance, other areas and fishing practice) where DSSs already resulted in an exemption to the landing obligation being granted; and reducing the observation periods in captive DSSs by supporting with vitality assessment at experimental closure, since the highest mortality is generally found during the first days of the observation.

### 4.2. Reduction: Methods That Minimize the Number of Fish Used per Experiment

The number of fish used in experiments is clearly a critical issue in STECF evaluations. However, STECF do not necessarily consider the statistical confidence in the estimates [36], and the assessment is based on expert subjectivity. Regulations stipulate the use of a minimum number of fish that ensures the most satisfactory results (Table 2, C). This requires a clear definition of high survival and an estimate of the statistical strength (power) of the estimate. The statistical strength basically depends on the sample size, the size of the desired effect, and the level of significance at which we want to detect the effect, which is generally defined as 95% in DSSs (as well as in other fisheries science areas).

The case studies presented in Table 1 demonstrate a variable number of fish assessed between experiments. We modeled the number of fish necessary to detect the limit for “high survival”. We used a non-parametric Kaplan–Meier model, since this is the most used method for modeling discard survival data [8]. It is important to note that it is the number of surviving fish that is relevant (not number assessed) for a scientific confidence estimate of final mortality, whereas present STECF practices evaluate the number of assessed fish. Our modeling is presented in Figure 1, together with the survival data in the DSSs presented in Table 1.

The lower 95% confidence limits (black dotted lines) reflect the potential limit for demonstrating “high survival rates” for estimated survival probabilities (blue dashed lines). For example, for a survival probability = 0.7 (70%), a minimum of around 70 surviving fish is required for the lower 95% confidence limit to be above 0.6 (blue X mark in Figure 1). Consequently, we can estimate the number of fish required for a given estimated survival probability to be statistically significantly higher than a limit (“high survival”). Increasing the number of surviving fish decreases the range of the 95% confidence level. If the estimated survival is around 90%, around 60 surviving fish would be required if the estimated survival probability needs to be significantly different from 80% survival, whereas more than around 250 fish would be required if the estimated survival is 85%. Since the curves tend to flatten when there are more than 100 surviving fish, 100 fish can be a useful guideline; however, prior knowledge about the expected survival rate when planning is necessary to reach this goal. Most of the DSSs presented in Table 1 have less than 100 surviving fish, and the lower 95% confidence limits are relatively wide (several around 20%). Since the number of control fish is also relatively low, this can add further uncertainty to estimates. Figure 1 is a useful guide that indicates the importance of including an assessment of 95% confidence limits in the assessment of the results (STECF) and when DSSs are designed.

### 4.3. Refinement: Methods That Minimize Fish Suffering

The duration of the capitative observation period might be required in the application for animal experimental permission (in our experience, the maximum approved is 10 days). ICES guidance does not specify a specific duration for the observation period when assessing discard survival, but it is recommended that monitoring be continued until mortality approaches an asymptote [7,8]. In practice, this is very often not the case because of experimental limitations, such as the availability of facilities, permission to conduct animal experiments, and the economic resources available. Survival is typically the lowest during the first few days after capture [7,8,43,44]. This trend is also observed in Table 1 and has also been observed in other DSSs [28,32]. Experiments with longer observation periods do not report any substantial changes in survival rates after a 10-day period or when reaching a 3-day plateau with full survival [28,31]. The studies in Table 1 demonstrate that mortality is still observed during the last period before experimental closure in most studies, and in several cases, it is observed at a level where that could influence the confidence of the final estimates.

We can observe several potential refinement practices that could be implemented to reduce the observation period: a clear definition of “high mortality” for study completion when the “high survival” limit is exceeded by the 95% confidence limit (guided by Figure 1); the use of controls to indicate unintended experimentally induced mortality; identifying the numbers of days without mortality necessary to reach or predict an asymptote; and support observations with verified vitality assessments at experimental closure that indicates future survival potential.

Storing facilities should consider fish wellbeing (Table 2, O), which is in harmony with reducing potential unintended and experimentally induced mortality. The use of substrate or refugee elements may satisfy ethological needs for some species, but should also be balanced against ensuring that the water is of high quality and the necessity for frequent individual visual inspections. The approval of fish storing facilities by animal research inspectorates is often mandatory.

One of the main challenges to overcome when assessing mortality is avoiding death as the endpoint (Table 1, I). Conventional definitions to determine whether a fish is dead can be the lack of respiration, the onset of rigor mortis, or a lack or color of gills [8], but do not fulfill this criteria (death as the endpoint) and moribund fish should be identified at an earlier stage. Therefore, species-specific signs of early moribundity must been identified, which is also requested for our own recent animal experimental permissions. Examples of relevant indicators could be a loss of equilibrium, unresponsiveness to a gentle nudge on the caudal peduncle, or changes in epidermal color. This highlights the necessity to understand these indicators better, in order to set an upper limit of pain, suffering, and distress (Table 1, K) and to find the critical point at which welfare is compromised and when the fish should be euthanized (Table 2, L). The identification of individual moribund fish requires the frequent monitoring of individual fish by an adequately competent staff (Table 2, Q). In previous DSSs, monitoring has typically been carried out every 24 h [8], and in the examples from Table 1, monitoring was carried out from 0- to 3- times per day. This is not always sufficient to identify moribund fish before a terminal endpoint. Monitoring should be particularly frequent in the first days of the experiment, when survival is typically the lowest. Our most recent AW permissions encourage hourly monitoring during the first period and 3–4 times per day throughout the rest of the study period. To carefully monitor individual fish, facilities that have been designed for this purpose are required, and cages that are anchored to the seafloor, which are used in some studies (Table 1), make it difficult or virtually impossible to regularly inspect and monitor individual fish.

The use of analgesics (pain relief), anesthetics (temporary loss of sensation), and euthanasia is essential for pain relief and suffering (Table 2, K). However, the information that is available to facilitate a humane choice is limited, and appropriate methods are interspecific [48,49,50,51]. Therefore, prior pre-experimental studies of available species-specific information are of importance in relation to the use of hypodermic needles (tagging or internal fluid sampling) or euthanasia for moribund fish or at the end of the experiment.

## 5. Discussion

AW regulations and ethical considerations are steadily becoming more prevalent in commercial fisheries and fisheries science. There are differences in the guidance, requirements, and regulations as well as in their interpretation among countries and scientific journals. However, it should be emphasized that even when procedures are permitted by AW regulations, there is room for ethical considerations to change experimental procedures in most cases. The 3Rs are now explicit in the legislation governing animal use for research in the E.U. and internationally, and the 3Rs principle is the basis for AW policy, a prerequisite for official approval for animal research DSSs, and the practice of modern research.

DSSs is a rapidly increasing research area with high political priority and many stakeholders, most of whom are not directly involved in the experimental procedures and have minimum knowledge on AW regulations. These stakeholders are the administrators of research funding agencies, the evaluators of research proposals, the politicians who are responsible for regulations, the mangers evaluating results in relation to the “high survival exemption”, and the reviewers and editors in scientific journals, and the fishing industry. Therefore, a better knowledge of basic AW regulations and basic ethics is of importance for more than researchers. There are some fundamental ethical juristic issues in relation to the landing obligation that should be considered in depth. The overall aim of the E.U. landing obligations is to improve resource utilization, while no considerations have been addressed to the constitutional basis of the European Union that stipulates full regard should be paid to fish welfare requirements. Therefore, fundamental basic ethical considerations regarding the level of pain, suffering, distress, or lasting harm caused by the on-deck handling of fish contra discard under a landing obligation should also be weighed before any decisions are made about the fate of any fish.

Having conducted fisheries research and DSSs, we are certainly aware that AW regulations are challenging, but the only option is to follow them and face the challenge head on. A comprehensive plan that takes AW considerations and regulations into account and that implements the 3R principles is mandatory. This also provides a very good opportunity to improve scientific quality and to go through experimental procedures and increase efficiency and long-term planning. Importantly, the 3R approach implies that satisfactory results are going to be obtained while avoiding the unnecessary loss of animals, which requires an optimized experimental design that is based on the development of a statistical evaluation and the use of evidence-based approaches.

Despite a comprehensive ICES guideline manual on DSS experimental procedures, there is a high variation between DSSs, something that was clearly demonstrated in the present study by a highly relevant case study (plaice). Furthermore, the ICES guideline manual does not address how to handle ethics and relevant AW legislation during experimental procedures. By studying the regulations and requirements, specific case studies, and examples of potential measures, we identified several considerations that should be taken into account when planning DSSs, which are summarized in Table 3. There are definitely many measures that can be considered when planning future experiments to improve AW. Additionally, there is an increasing number of DSSs that have been published more generally in relation to fish welfare, and the literature should be updated continuously in order to ease the planning process. It is essential to highlight the importance of sharing information. This can be carried out by adding supplementary information in papers, which is optional in most journals, or reference numbers for publicly available animal experiment permissions.

Completing a STECFs assessment for scientific evidence to prove high survival rates is challenging. A clear definition of “scientific evidence” and “high survival” would not only be a solid framework through which researchers could conduct more focused DSSs, but it would also improve experimental planning in relation to AW regulations. With the increasing amount of DSSs across fisheries and species, a broader assessment should be possible rather than the expectation that a new DSS be conducted for every new case, which would eliminate the necessity for many studies.

## 6. Conclusions

AW regulations and ethical considerations are steadily becoming more prevalent in research areas related to commercial fisheries. Because of the E.U. landing obligation, discard survival is a rapidly increasing research area. DSSs are “frontrunners” in fisheries science research areas that are embedded by animal research welfare requirements. AW regulations are of high importance for experimental planning and all DSSs are embraced by the same E.U. regulations. Nevertheless, knowledge of basic AW regulations and principles is limited and there is a lack of guidance for researchers.

We identified the relevant paragraphs for DSSs in E.U. regulations. E.U. regulations are built on internationally established principles, particularly the 3Rs. The 3Rs principles are now the basis for AW policy in the E.U. and internationally, and the practice of modern research. We investigated experimental procedures by bringing in relevant examples, using output results, and relating this information to relevant AW guidelines and regulations by focusing on implementing the 3Rs principle. We identified many measures that can be considered when planning future experiments to improve AW. This also provides a very good opportunity to improve scientific quality and to go through experimental procedures and increase efficiency and long-term planning. We highlighted the importance of the sharing of information and a greater focus on AW in relation to conducting DSSs, but also for other fisheries science research areas.

It is certainly a challenge to assess mortality without reaching a terminal endpoint. However, we highlighted several examples and considerations concerning how to follow the 3Rs when planning DSSs. This is only a starting point and not a final list.

## Figures and Tables

**Figure 1 animals-12-00782-f001:**
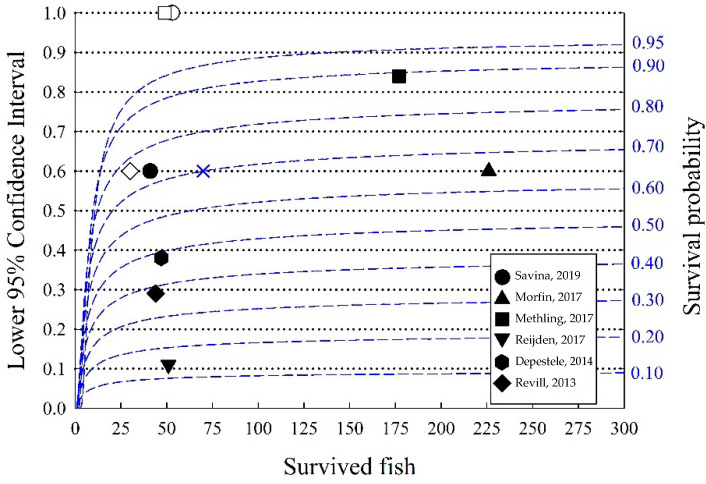
The plot shows the lower 95% confidence limit (left axis and dotted lines) for estimated survival probabilities (right axis with blue numbers and blue dashed lines) using the Kaplan–Meier model as a function of the surviving fish. Black symbols represent the studies presented in Table 1, and white symbols represent control groups (when available). The blue cross was used as an example in the test.

**Table 1 animals-12-00782-t001:** Overview of discard DSSs on plaice captured during the winter season (1 November–30 April), including information on gear type (bottom otter trawl, OTB; pulse trawl, PT; beam trawl, BMT); fishing area; methods (land-based tanks (tanks) or seabed cages (cages)); inspections of the captative fish (no/day); vitality assessments conducted (before indicates before capitative observations); and number of hauls. Days indicate the length of the captive observation period, and the number of assessed fish (start of captive observation period) and number of surviving fish (start of captive observation period) are also represented. High early survival indicates whether survival rates were the highest in the first 3 days, asymptote days indicate the number of days with 0 mortalities prior to the end of observation periods (asymptote) and if the decline in mortality over the last 0 days of observation was below 5 percentage points (*), and finally, the table depicts survival at the end of the observation period.

Gear–Area [Reference]	Methods	Inspection (No./Day)	Vitality	Hauls	Days	Assessed (No.)	Surviving (No.)	High Early Survival	Asymptote (Days)	Survival
OTB–Skagerrak [33]	Tanks	3	Before	6	14	54	41	No	<1	75%
OTB–English Chan. [30]	Tanks	2	Before	17	3–6	348	226	Yes	<1	63–67%
OTB–Skagerrak [29]	Tanks	1–2	Before	5	10	199	177	Yes	<1 *	89%
PT–North Sea [31]	Tanks	1–2	Before	7	21	349	51	Yes	2–3 *	15%
BMT–North Sea [27]	Tanks	2	No	5	3	97	47	Yes	<1	48%
BMT–English Chan. [26]	Tanks	1	No	<7	3	80	58	Yes	<1	37%
OTB–Baltic Sea [37]	Cages	None	Before	6	5–7	226	NA	NA	NA	~30–95%

**Table 2 animals-12-00782-t002:** Main regulations extracted from the E.U. Directive 2010/63/EU that are of relevance to DSSs.

Relevant Passages
Included: vertebrate animals, including cyclostomes (lampreys and hagfishes) and cephalopods. Animal research or education definition is any invasive or non-invasive animals that leads to a level of pain, suffering, distress, or lasting harm that is equivalent to or higher than an injection with a needle (known as the needle criteria). Members States are allowed flexibility to maintain national rules aimed at more extensive animal protection. Choice of methods should ensure the most satisfactory results that are likely to cause the minimum amount of pain, suffering, or distress and the use of the minimum number of animals. Care and use are governed by internationally established principles of replacement, reduction, and refinement (3Rs) that should be considered systematically. Comprehensive project evaluation taking into account ethical considerations and implementation of the 3R principles should form the core of an authorization/license. Animal use must be restricted to areas that may ultimately benefit human or animal health or the environment. The selected methods should avoid death due to severe suffering before reaching death as an endpoint as much as possible. Humane endpoints should ensure euthanasia before spontaneous death. Other methods to replace the use of live animals are desirable and should be used when possible. The general or local use of anesthesia and analgesia should ensure that pain, suffering, and distress are kept to a minimum. In general, experiments must be performed under anesthesia. There should be an upper limit of pain, suffering, and distress. The most appropriate decision should be made regarding the future of the animals, and animals should be killed if welfare is compromised. Killing should be carried out by a competent (trained) person using methods that are appropriate for the species. Accommodation shall ensure that an animal can satisfy physiological and ethological needs and that any defect or avoidable pain, suffering, distress, or lasting harm discovered is eliminated as quickly as possible. Transportation shall be carried out under appropriate conditions. Animals taken from the wild shall not be used, except if the study cannot be performed without them. Staff should be adequately educated, trained, and competent. This includes mandatory courses in laboratory animal science. Objective information concerning projects using live animals should be made publicly available.

**Table 3 animals-12-00782-t003:** Considerations when conducting DSSs.

-Planning	Considerations
Overall aims	To estimate “high survival” (above a defined limit) or more general discard survival mortality rate. To estimate short-term or long-term survival (define length of period). Definition of “high survival”. Definition of “scientific evidence”.
Regulations and requirements	Early dialogue with national authorities is important (permission often takes time to attain). Potential experimental designs not requiring or conflicting with any AW regulations. Assess how to address E.U. directives. Assess national regulations. Review similar studies for experimental procedures. Study relevant animal experimental permissions. Consider scientific journal requirements for relevant journals. Use the prepared guidelines while planning the experiment (PREPARE (www.norecopa.no; accessed on 16 March 2022)) or other relevant guidance documents.
Replacement	Assess the relevance of the study based on prior STECF evaluations and DSSs that have been conducted or that are in progress. How to meet the required “scientific evidence” for STECF. Assessment based on available information alone. Relevant alternatives for captive observations, such as RAMP monitoring.
Reduction	Define “high mortality” to terminate the experiment when the limit is reached. Standardize experimental studies. Assess the number of fish required (statistical power calculation). Identify the criteria that can reduce the length of the observation period. Good storage facilities that fulfil fish needs and required inspections. Plan frequent individual inspections at a frequency according to mortality rate. Use experienced, educated staff.
Refinement	Define moribundity as part of the humane endpoints. Consider procedures and use of analgesics, anesthetics, and euthanasia.
Reporting	Provide the application number for AW permission. Make AW permission public. Write a detailed ethical statement section. Make sure to include all the relevant information for other groups to reproduce the results (ARRIVA guidelines). Provide details as supplementary information in scientific journals.
Other considerations	Broader considerations of AW in relation to capture process and discards. Staff education and training. Potential presence of national 3Rs centers.

## Data Availability

Data reviewed in this Commentary can be found in the indicated references.

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
