# Peer review of "Estimating Discard Mortality in Commercial Fisheries without Fish Dying: A 3R Challenge"

_animals, 2022, doi:10.3390/ani12060782_

Round 1

Reviewer 1 Report

As the authors mentioned, in Fisheries Science and particularly in dealing with discards of animals with high survival rates, animal welfare is a challenge. Thus, the authors focused on AW regulations in discard survival studies, using an approach essentially pragmatic, and emphasizing commercial fisheries of the European Union. 
The experimental procedures investigated are helpful examples and the outputs associated with relevant AW issues, highlighting the implementation of the 3R principles.
Although the focus is on EU regulations, the paper is notable in covering fish welfare regulations. In addition, the article will be an essential citation source in the growing discard survival studies, particularly improving experimental planning regarding AW regulations. Congratulations to the authors for the guidelines about this important topic.

Specific

A glossary is required for rapid consultation of a lot of terminology and acronyms.

Author Response

Thank you for the comments. 

A glossary is a good point, this will help researchers not into the research area.

Action: we added a glossary as an Appendix, but mentioned in the introduction.

Reviewer 2 Report

This paper is about the relationship between the EU’s landing obligation (discard ban) and the EU’s animal welfare regulations. The authors argue that when the landing obligation was introduced in 2015, it permitted discards of species that had a high chance of surviving, but it did not relate this exemption to animal welfare regulations:  

the guideline does not address how to handle animal welfare regulations, although this is becoming of increasingly importance when planning discard survival studies” (p 2 section 1.2)

More specifically, the authors focus on the failure to factor animal welfare considerations into scientific investigations into discard survival rates.  

In my view, this is a very good paper which deserves to be published virtually as it stands because it contributes important insights to a new and serious issue – of reconciling the ban on discards with increasingly stringent animal welfare standards.

However, I have one suggestion for the improvement of the paper. Can the authors clarify whether they are more focused on (1) the animal welfare implications of scientific research on the discard survival rates of fish species (e.g., how much distress is caused to fish by laboratory experiments conducted on them to see how long they survive in different conditions); or (2) the animal welfare implications of releasing discarded fish into the sea (e.g., whether it may be better for a fish to die quickly on deck rather than risk suffering a long, lingering death when thrown back into the sea)?   

Author Response

Thank you for the comments. 

Reviewer: However, I have one suggestion for the improvement of the paper. Can the authors clarify whether they are more focused on (1) the animal welfare implications of scientific research on the discard survival rates of fish species (e.g., how much distress is caused to fish by laboratory experiments conducted on them to see how long they survive in different conditions); or (2) the animal welfare implications of releasing discarded fish into the sea (e.g., whether it may be better for a fish to die quickly on deck rather than risk suffering a long, lingering death when thrown back into the sea)?   

Our comments: it is a very good point. We do not have the answer, but certainly find that it should be carefully addressed and relevant to mention here. Therefore we address in the discussion by following text:

The overall aim of the EU landing obligations is to improve resource utilization, while no considerations have been addressed to the constitutional basis of the European Union that stipulates full regard should be paid to fish welfare requirements. Therefore, fundamental basic ethical considerations regarding the level of pain, suffering, distress, or lasting harm caused by the on-deck handling of fish contra discard under a landing obligation should also be weighed before any decisions are made about the fate of any fish.

Additionally we got professional English editing service for language improvements.